# Improving Fatty Acid Profile in Native Breed Pigs Using Dietary Strategies: A Review

**DOI:** 10.3390/ani13101696

**Published:** 2023-05-19

**Authors:** Ainhoa Sarmiento-García, Ceferina Vieira-Aller

**Affiliations:** 1Área de Producción Animal, Departamento de Construcción y Agronomía, Facultad de Agricultura y Ciencias Ambientales, Universidad de Salamanca, Av. de Filiberto Villalobos 119, 37007 Salamanca, Spain; 2Estación Tecnológica de la Carne, Instituto Tecnológico Agrario de Castilla y León (ITACyL), Calle Filiberto Villalobos 5, 37770 Guijuelo, Spain; vieallce@itacyl.es

**Keywords:** autochthonous pork, fatty acid profile, healthy meat, pork quality

## Abstract

**Simple Summary:**

Over the last decade, intensive pig-rearing systems have been questioned due to the farming method, leading to increased consumer demand for environmentally and animal-friendly production. These production systems, among other particularities, involve raising indigenous breeds that are widely appreciated for their rusticity and the high quality of their meat. These advantages are related to its meat’s intramuscular fat content and lipid profile, which improves technological and sensorial properties and promotes healthy products. Despite genetics’ influence, dietary composition plays an important role in meat quality. Many researchers have focused on evaluating the different dietary strategies for improving the quality of meat. In this sense, several natural sources are available as potential feed ingredients for pigs. The current review evaluated the studies about improving the meat quality of native pigs using dietary strategies. Due to the increased relevance of these breeds, further studies on this topic are required, as they are scarce compared to commercial breeds.

**Abstract:**

Meat from native-bred animals is growing in popularity worldwide due to consumers’ perception of its higher quality than meat from industrial farms. The improvement in indigenous pork has been related to increased intramuscular and unsaturated fat and a reduced saturated fat content resulting in a healthy product with enhanced sensorial attributes. This manuscript aims to provide an overview offering useful information about the fat content and the fatty acid profile of different autochthonous pork. Fat content and fatty acid profile are greater in native than in industrial pig breeds, even though certain factors, such as genetics, nutrition, farming system, age, or slaughter weight, may influence these variations. Among that, studies on dietary strategies to improve these parameters have been evaluated. According to the results obtained, many natural ingredients could have a positive effect on the lipid profile when added to indigenous pigs’ diets. This fact may promote autochthonous pork intake. Nevertheless, there is a wide range of potential natural ingredients to be added to the indigenous pig diet that needs to be evaluated.

## 1. Introduction

Pork is one of the most popular and traditional meats worldwide, composing about 33% of the total meat intake [1]. Its consumption has a wide variety, as it can be eaten raw, cured, or cooked [2,3,4,5]. About 56% of the world’s pork is produced in Asia, where China (with 48%) is the leading producer. This is followed by North America and the European Union, which represent 12% and 21% of world pork production, respectively. In Europe, Germany, Spain, and France are the main producers of pork [6]. A substantial amount of pork farming is provided by intensive production systems, where commercial hybrids are kept confined in buildings on slatted floors, leading to a reduction in total production costs [6]. According to industrial criteria, commercial hybrid pigs enhance their productive performance due to their beneficial high feed conversion efficiency, fast growth rate, and economic benefits [7]. However, in recent years intensive systems have been questioned [8], and several studies have revealed an increasing demand and preference by consumers for alternative systems that are based on environment and animal-friendly methods. These new systems include different aspects, such as housing conditions (i.e., outdoor access, free-range), feeding with natural resources, or the use of autochthonous/native breeds. Autochthonous breeds are well adapted to the environmental conditions, with higher resistance to disease and improved utilization of pasture than commercial hybrids. In addition, the intake of small shrubs and other vegetative species by indigenous pig breeds, combined with ruminant livestock, contributes to their removal and the prevention of fire prevalence [8,9]. Rearing autochthonous or indigenous breeds have been associated by consumers with eco-friendly farming practices, animal welfare, and a positive influence on meat quality [1,8,9,10,11]. 

Despite the advantages of commercial hybrid in performance production, meat from hybrid pigs is usually typified with poor quality attributes, including high drip loss, color loss, and decreased intramuscular fat content (IMF), making this meat undesirable for transformation into processed products [1]. The differences in meat quality between hybrids and autochthonous pigs are related, especially to their IMF and fatty acid profile [7], which affect sensory attributes (such as juiciness or tenderness) and processing properties [9]. The commercial swine industry employs a limited number of breeds in crossbreeding to benefit from heterosis effects, while there are wide indigenous pig breeds spread worldwide [6] (Figure 1). Many indigenous pig breeds are well-known for their rusticity. The term “rustic” is used to describe conventionally raised, slow-growing, local breed pigs, usually slaughtered at an advanced age, which provide carcasses and meat with high pigmentation of muscle and fat [12]. Among these, meat from different breeds, such as the Iberian [3,13,14] Celta (northern Spain) [15,16], Chato Murciano (south-eastern Spain) [17,18], Bisaro (northern Portugal) [1,19], or Cinta Senese (North of Italy) [20,21], have been described by previous authors. 

Despite the important role of genetics on the quality and sensory acceptability of meat [9,11], the diet has been described as one of the most crucial aspects which could modify the content and composition of fat [2,3,22] contributing to the development of healthy meats [23]. Hence, the diet (and, in particular, the dietary fatty acid supply) is a key factor for modulating pork meat’s fat content [3] and, in addition, could enhance performance production (reducing feed intake and increasing feed efficiency [24]). The efficacy depends on the type of fat or oil added, the time of feeding, the amount included, and even the tissue assessed [24]. Several studies have described how the lipid deposits of pigs are changed as a function of pig nutrition [3,13,20,25,26,27]. Therefore, these strategies would seem to be of more relevance to obtaining high-quality products derived from autochthonous breeds, such as those protected by brands, such as Protected Designation of Origin (PDO) or Protected Geographical Indication (PGI) [28]. These brands perform an important role in marketing indigenous pork, so improving quality opens up niche markets for this meat. 

It is not easy to compare the available information about meat quality from different autochthonous pigs due to the great variety of breeds. The information shown below should be carefully evaluated due to the different management practices, the high variability among breeds, and even the high variability within the breeds [7]. However, the results described below may help get an overall insight into these breeds’ meat quality (in terms of the IMF and the fatty acid profile) and their differences from industrial hybrids. Moreover, concerning the importance of meat quality from these breeds could be interesting to meet the strategies available to enhance their attributes. Therefore, the focus of this review was to summarize the latest information available for the characterization of the fatty acid profile from different autochthonous pork and the main feeding strategies to improve their fatty acid profile. In this regard, ingredients that require minimum treatment to be included in the diet of autochthonous pigs have been reviewed.

## 2. Factors Affecting Intramuscular Fat Content (IMF) and Fatty Acid Profile

In mammals, medium (FM) and short (FS) fatty acids (with aliphatic tails fewer than 13 carbon) are absorbed faster than long fatty (FL) acids. FL, which have aliphatic tails longer than 12 carbons, are slowly absorbed and metabolized in the organism, accumulating quickly in the liver and adipose tissue. FL requires fatty acid transport carriers to enter the mucosal cells of the small intestine and can be used to synthesize TAGs. Fat storage in animals results from an unbalance of synthesis and degradation. If fatty acid synthesis exceeds consumption, fatty acids are accumulated in the cells rather than be mobilized to supply energy. Fatty acid synthesis can be divided into endogenous and exogenous pathways. Endogenous fatty acids are lipids synthesized de novo from intracellular acetyl-CoA. The primary sources of intracellular acetyl-CoA are aerobic carbohydrate oxidation, the β-oxidation of fatty acids, the catabolism of specific amino acids, and the oxidative degradation of ketone bodies [29]. The primary end product of this fatty acid synthesis pathway is palmitic acid (16:0), which can be extended to stearic acid (C18:0). Both palmitic acid (C16:0) and stearic acid (C18:0) can be transformed into MUFA, giving palmitoleic acid (16:1 n-9), and oleic acid (18:1 n-9), respectively, due to the ∆9-desaturase activity [29]. In this sense, SFA presented in pork meat is mostly obtained from synthesis de novo from dietary carbohydrates, while PUFA comes from the diet as the animals cannot synthesize them, and MUFA can be obtained in these two ways [30]. Therefore, nutrition has a pivotal role in the fatty acid profile. The exogenous fatty acids are derived mainly from energy and carbohydrate intake, which are primarily present in circulation [30]. They may also be absorbed, digested, and used for TAG and phospholipid synthesis [29]. Energy and carbohydrate intake perform a principal role in de novo fatty acid synthesis, whereas lipid-rich diets inactivate endogenous lipogenesis. A rise in glucose metabolism is necessary for most lipogenic enzymes to prompt transcription. Nevertheless, in vitro studies have demonstrated that other precursors, including acetate, lactate, citrate, and glycerol, can promote pig lipogenesis [30]. The important role of nutrition in the IMF and the fatty acid profile have been described by different authors [4,29]. However, it is important to point out that its absorption and distribution are strongly determined by genetics [9,30]. For example, it has been described that Iberian pigs have a greater capacity to synthesize or store MUFA, which may be due to a higher ∆9-desaturase activity than hybrids pigs [30]. Li et al. [31] reported that the presence of the FAT1 gene from Caenorhabditis elegans can significantly increase the level of PUFA and decrease the ratio of n-6/n-3 PUFA in pigs. The review carried out by Pugliese et al. [7] described that native pigs show a high predisposition to deposit MUFA (mainly oleic acid), while hybrid pigs show an increased level of SFA. Authors attributed those differences as a consequence of novo lipid synthesis between breeds.

Nevertheless, previous studies suggest that lipid profile differences could be detected even if the breed is the same; therefore, additional factors must be considered, such as the muscle examined and the age or weight of the slaughter [17]. For example, it has been described that “red” muscles would have a greater percentage of PUFA than white muscles due to a higher proportion of phospholipids (*longissimus* vs. *gluteobiceps*) [23,32]. In the same line, Turner et al. [33] described an increase in the total fat content (from 2.9 to 22.6%) of *longissimus* by adding various cuts to commercially trimmed pork chops, affecting the lipid profile and increasing the SFA content. Regarding weight and age, Garrido et al. [34] proposed that during the last rearing period (about 90 kg), the differences between breeds are more evident as body weight and fat deposition increase. In this sense, Pugliese et al. [7] pointed out that the capacity of native breeds to deposit MUFA could increase with age. 

As described above, the IMF and the lipid profile of pigs are strongly influenced by several factors, so possible variations among the studies presented below should be considered. The lipid composition of native pork and its advantages over industrial pork are described. Considering these advantages, it is useful to know which nutritional strategies could enhance the lipid profile that could lead to an increase in the intake of native pork.

## 3. Indigenous Pork Fat

### 3.1. Intramuscular Fat

IMF has a pivotal role in the sensorial attributes of meat and contributes to some aspects of meat health status. Assessing IMF would appear as the best method for distinguishing indigenous from commercial pork [7]. Cebulska et al. [35] demonstrated a higher IMF in the Puławska pork meat (an indigenous Polish pig) than in their crossbreeds (4.07% vs. 1.70%), which is in accordance with Kušec et al. [28]. Similarly, Franco et al. described [15] a reduction in IMF from the Celta breed (5.22%) (a north Spanish pig) to the cross between Celta and Duroc (3.96%) and the cross of Celta and Landrace (3.08%). The studies reviewed indicated a variable IMF percentage, which decreases as the animal’s genetic selection increases. All breeds reviewed (i.e., Cinta Senese, Celta, Mangalica, or Prestice Black-Pied) [9,15,16,21,36] have shown IMF ranges from 3 to 10%, with the highest levels in the Iberian pig (an autochthonous Spanish pig) obtaining more than 10%. Nevertheless, these values could be affected by the rearing system. For example, in the study carried out by Vieira et al. [3], where the IMF of *Longissimus thoracic et* lumborum muscle from Iberian pigs reared under intensive conditions, IMF ranged from 8.14% to 11.53%. These values could be lower than previous authors due to the intensive conditions. 

It has been considered that a minimum level of IMF (2–2.5%) may be acceptable for obtaining satisfactory meat sensory traits. The current results show that native genetics lead to an improvement in the quality of pork than commercial hybrids [7,28]. These results are probably due to the genetics of the native breeds, which have been linked with changes in the absorption and deposition of fat [28]. Moreover, the rearing system could affect these variations [17]. Furthermore, this increased susceptibility to fat tissue accumulation rises at high slaughter weights when the proportion of adipose tissue increases and the lean tissue decreases [28], which is in accordance with indigenous pig rearing. Auqui et al. [17] reported that IMF varied from 5.63 to 7.01% in Chato Murciano, reaching the maximum value as the age increased. The results showed in the current review demonstrated that indigenous pork’s IMF content is higher than commercial hybrids, enhancing the sensory attributes of the meat. In this regard, crossbreeding native pigs with commercial hybrids, such as Duroc, Yorkshire, and Landrace, whose IMF content is less than 2.5%, is a suitable approach for improving the IMF of these crossbreeds [37]. It would be an attractive approach for increasing the use of indigenous breeds even under intensive farming systems.

### 3.2. Fatty Acid Profile

The lipid composition of pork includes 568 different lipid species with 139 triglycerides (TAG). There are three types of fatty acids in TAG, including saturated fatty acids (SFA), monounsaturated fatty acids (MUFA), and polyunsaturated fatty acids (PUFA) [29]. According to the current health recommendations as proposed by EFSA [38], the diet should be rich in PUFA and MUFA and low in SFA due to the apparent relationship between plasma cholesterol and cardiovascular disease [7,39]. In pork, IMF ranged between 2% to 15%, which constitutes about 35–40% of SFA, being C14:0 as the major fatty acid content. 

Based on the above information, many recommendations have been proposed to reduce the intake of red meat, such as pork. In response, several studies have focused their efforts on examining the content and composition of pork fat, and different strategies to modify it. These findings have shown that composition and the deposition of fat in pork could vary with genetics, with important differences between commercial and indigenous pigs. For example, the experiment carried out by Nevrkla et al. [9] with Prestice Black-Pied (an indigenous Czech breed) showed a decrease in SFA (C8:0, C10:0, C15:0, C22:0) and an increase in MUFA (mainly in C18:1 n-9) content which is consistent with the findings of Tomović et al. [36] for white Mangalica pigs (an indigenous Hungarian pig). In the same line, Kušec et al. [28] compared Black Slavonian barrows with commercial hybrids, and they observed a decrease in palmitic acid (C16:0) and an increase in beneficial unsaturated fatty acids, such as palmitoleic acid (C16:1 n-7), oleic acid (C18:1 n-9), gamma-linoleic acid (C18:3 n-6), and nervonic acid (24:1 n-9), which increase MUFA and PUFA n-3 content. Similar trends have been reported by Serra et al. [21], who have shown higher MUFA content in Cinta Senese (an autochthonous pig from Tuscany, Italy) than in industrial pigs (Large White). Teixeira and Rodrigues [40] described higher contents of MUFA (C18:1 n-9, C16:0, and C11:1 n-7) in the Preto Alentejano breed (a South-Portuguese pig) than in commercial hybrid while inconclusive differences were found in PUFA. Related tendencies have been described by Lebret et al. [41], who showed an increase in content and a decrease in SFA for Basque Pork (nonselected, local French) compared to Large White (hybrid pig) Which coincides with those described for Iberian pork [42].

Not only are the PUFA, MUFA, and SFA content important to obtain a healthy product, the ratio of n-6/n-3 has been known to have nutritional significance [33,39,43], especially relative to the content of eicosapentaenoic acid (EPA), docosapentaenoic acid (DPA), and docosahexaenoic acid (DHA) [44], being necessary for the equilibrated biosynthesis of eicosanoids in the organism. Unbalanced n-6/n-3 ratios, which should be less than either 4:1 or 5:1 [44], have been linked to many disorders, including cardiovascular and inflammatory diseases or diabetes [23]. Furman et al. [45] showed a better n-6/n-3 ratio for Krškopolje (a Slovenian indigenous pig breed) than in hybrid pork, which is consistent with the findings observed by Nevrkla et al. [9] and Kušec et al. [28] for Prestice Black-Pied pork and Black Slavonian, respectively. Table 1 summarizes all studies described before.

Moreover, atherogenic indices (AI) and thrombogenic indexes (TI) have been described as important in identifying healthy products. Those values correspond to the quantities of particular SFA, MUFA, and PUFA from the n-3 and n-6 series. A lower AI and TI value demonstrate a decrease in SFA to unsaturated acids [19,35]. These values suggest a role in preventing or promoting pathologic processes in humans, including atheroma and/or thrombus development [46]. Lower IA (<0.5) and TI (1.2) values have been documented in indigenous breeds than in commercial hybrids [1,35], while no differences were reported in the research carried out by Cebulska et al. [35] with similar conditions. 

It may be said that overall, the fatty acid profile of autochthonous pork would seem optimal compared to those of commercial pigs, which in general, have a lower SFA and higher PUFA and MUFA content. These enhancements may lead to healthier meat with improved sensory attributes than commercial crossbred. Nevertheless, the latest outcome of a meta-analysis study indicates that the intake of SFA has no association with cardiovascular disease risk [47]. Thus, the recommendations for substituting red meat with other protein sources in the human diet have been reconsidered. Including indigenous pork meat in the human diet could be an adequate strategy over commercial hybrids [39].

## 4. Strategies to Improve the Fatty Acid Profile of Autochthonous Pork Meat

As noted above, for years meat intake has been linked with adverse health problems, due to the content of SFA [23], which has been associated with an increased prevalence of cardiovascular disease and various medical disorders [48]. Nevertheless, the fat content and the fatty acid profile perform an important role in the meat’s technological properties and sensorial attributes [23]. Thus, a growing interest in pig industries has been focused on the strategies for improving the fatty acid profile of pork meat. Several studies have suggested that the fatty acid profile of meat can be modulated by different mechanisms, including animal nutrition as the most effective way, and could interact with pig genetics [11]. Therefore, nutritionists are focusing on animal feed, especially in ingredients with high amounts of PUFA, for their consequent benefits to animal health and meat quality [49]. 

According to what was described before, dietary fatty acids are little changed in digestion and absorption in monogastric species than in ruminant species, and PUFA is deposited in pork tissues without biochemical changes [6]. Therefore, the fatty acid content of tissues reflects the fatty acid composition of the diet [50]. Nevertheless, efficacy for deposition in meat and subcutaneous backfat depends on the fat supplied (content and structure) and physiological pathways, such as de novo synthesis of fatty acids, rate of interconversion into different fatty acids and their metabolites and oxidation processes for energy use [3,24]. In addition, it is important to note that the starch content of the diet performs a significant role in modulating the deposition of fatty acids in muscle. Diets richer in fiber may affect lipogenesis, likely as a result of reduced available energy or the inhibitory effect of certain compounds, such as propionate [30].

Several natural sources could be potentially used as ingredients in animal feed due to their effect on the meat. Among these, plants, including seeds and their oils, algae, or insects, are a good source of PUFA and have been described as an adequate ingredient in animal diets, contributing to energy requirements and efficient production [18,34,51,52,53,54]. In addition, by-products generated during the processing of agriculture offer the possibility to generate value-added products, such as animal feed, that would reduce farming costs in a circular economy [52,53,55,56]. Most of the studies available have been carried out in hybrids pigs; nevertheless, according to the previous information, it could be more interesting to investigate nutritional strategies in autochthonous pigs to enhance their meat. This section summarizes the main findings of previous reports evaluating dietary strategies to modify the fatty acid profile of native pigs using natural dietary sources.

### 4.1. Acorns

Acorns are considered the most important feed ingredient for several native pig breeds (such as Alentejana, Bisaro, or Iberian) raised under extensively traditional conditions [1,8,26]. In this rearing system, known as *montanera*, acorns are used to fatten the pigs to provide the pig’s muscle and adipose tissues with typical chemical characteristics. Due to the importance of acorns during the *montanera*, numerous experimental studies have been carried out on the effect of an acorn-based diet on pork characteristics. Acorns from the evergreen oak (*Q. ilex rotundifolia*) come in a wide variety of shapes, sizes, and compositions among different trees. Acorns are a rich source of fat (more than 10%) and have other compounds, such as α-tocopherol, γ-tocopherol, or tannins, contributing to preventing lipid peroxidation. In addition, the planting of oaks for acorns favors ecological and socioeconomic possibilities, particularly in less favored areas [26]. 

Regarding the effect of *montanera* rearing systems on the fatty acid composition of Iberian pigs, Cava et al. [57] reported an increase in MUFA content (specially C18:1), a decrease in SFA content, and no difference in the content of PUFAs when Iberian “Montanera” was compared with Cebo Iberian pigs (concentrate-based diet). Similar results have been reported by Rey et al. [26], who described that meat from Iberian pigs which have been received acorns had an increase in oleic acid (C18:1 n-9), around 53–54%, and very low proportions of palmitic (C16:0) and stearic (C18:0) acids. Nevertheless, the improvement in the fatty acid profile related to acorn intake has been evaluated in other breeds. For example, Szyndler-Nędza et al. [8] pointed out that in Złotnicka pigs that had access to the acorns, the meat profile had been improved. Those authors showed a reduction of TI due to a lower SFA content (including C14:0) and a higher content of MUFA (particularly C18:1 n-7). Moreover, those authors described a lower susceptibility to peroxidation which is due to the content of MUFA, which are more resistant to peroxidation than PUFA. Some authors suggested that these improvements are due to a synergistic effect between pasture and acorn intake due to the extensive rearing, where the activity of the ∆9 desaturase and elongase enzyme increase. This fact could increase other fatty acids, such as C22:5 or C22:6, in the intramuscular fat. Additionally, as is mentioned above, acorns are rich in antioxidant compounds, such as α and γ-tocopherol, and tannins, to prevent lipid oxidation [26]. 

It could be summarized that acorn intake enhances the fat content and the fatty acid profile, which acts as a carrier for the flavor and, indirectly, juiciness [8]. However, the seasonal availability of acorns results in the fact that Iberian pigs kept under these conditions do not remain available all year round. Thus, several studies have been conducted to find alternatives to the use of acorns and to obtain a meat product with similar attributes [3].

### 4.2. Crude Olives and By-Products

Crude olives and their by-products have high levels of oleic acid (C18:1 n-9), making them an interesting ingredient in animal diets. Supplementation with crude olives and its by-products have been linked to an improvement in the meat fatty acid profile with an increase in the percentage of oleic acid and a decrease in SFA, without impairing growth rates [19,20,25,54,58]. Moreover, these products are a good source of certain antioxidant ingredients, such as polyphenols and tocopherols, which could decrease meat spoilage [20]. Thus, using by-products as animals feed could be a good strategy to decrease the costs and the environmental impact, which is in accordance with the autochthonous rearing procedures [53,54]. 

Several studies about the use of crude olives and by-products as ingredients in pig diet have been described. In the study conducted by Leite et al. [19] with Bisaro pigs fed with crude olive oil, an increase in IMF was observed concerning the control group, while in the study carried out by Palma et al. [53] in Iberian pork, no differences were reported among the control and the experimental group when olive cake oil was added to the diet. According to Garrido et al. [34], physical activity increases when animals are raised outdoors, leading to a high energy intake and a low-fat deposition. For the fatty acid profile, Bisaro pigs fed with olive cakes recorded lower values for some SFA, such as C14:0, C16:0, C18:0, and total SFA [19]. Moreover, these authors described an increase in C18:1, C18:2, C18:3, and PUFA content. Similar trends have been described by Liotta et al. [58] in Pietrain pork; these authors demonstrated that the inclusion of 50 g/kg of olive cake and 100 g/kg of olive cake reduced the SFA content by 38.71% and 34.62%, respectively. These results agree with those obtained by Serra et al. [20], who described a healthy sausage (high PUFA and low SFA content) from Cinta Senese when olive pomace was added to the diet. According to Yi et al. [29], these enhancements could be attributed to the findings that increasing high oleic acid in the diet reduces the expression of fatty acid synthase, reducing fat synthesized de novo via the endogenous lipogenic pathway.

From the standpoint of health and technological properties, it is preferable to reduce SFA and increase PUFA and MUFA [38], but increasing PUFA would raise the susceptibility to oxidation in meat products, which could affect flavor and pork’s nutritional value [24]. Thus, to solve this problem, antioxidant supplementation of the pig diet is strongly recommended when products rich in oleic are added to the diet [6]. Based on these findings and considering that olives are widely used and the number of by-products that are generated, including these ingredients in the diet of indigenous pigs, could be considered an optimal dietary strategy to reduce feed costs and enhance the lipid profile.

### 4.3. Chestnuts

Chestnut (*Castanea sativa Mill*) belongs to the *Fagaceae* family which is popular in South Europe and Asia. The chestnut tree has a major impact on the economy of several regions, being the main chestnut forests of Europe located in France, Italy, and Spain. Chestnut fruits are extensively used for both human and animal intake, as they are a rich carbohydrate source. Chestnut fruits are underutilized, particularly the small chestnut trees and industrial by-products, although their chemical composition provides an interesting approach to use as an ingredient in livestock feed. 

Temperan et al. [59], in Celta pigs, described that IMF was slightly increased in pigs feeding chestnuts than in control, which is in accordance with the results obtained by Domínguez et al. [52]. Moreover, Temperan et al. [59] demonstrated that, although these values were not significant, chestnut feeding reduces SFA and increases MUFA in *Longissimus dorsi*. Bermudez et al. [55] reported the increase in chestnut in the finishing diet could lead to an increase in the MUFA and a reduction in the SFA content in the *Biceps femoris* of the dry-cured ham. The increase in MUFA is closely linked to the linoleic (C18:2 n-6) and α-linolenic (C18:3 n-3) content of the tissues, which are dependent on the dietary content since these acids cannot be synthesized in the tissues [3,52]. Thus, a rising linoleic and α-linolenic content in muscle fat can be expected as the proportion of chestnuts in the diet increases. Moreover, these authors described a healthy fatty acid profile linked to the low n-6/n-3 ratio observed in pigs feeding with chestnuts. Similar results have been described by Pugliese et al. [51] in Cinta Sense. These authors proposed that chestnut feeding resulted in high adipose tissue unsaturation, particularly concerning PUFA n-3 and n-6. Based on the above findings, chestnuts could be considered a good source of fatty acid profile and an interesting option for autochthonous pig feedings as an alternative way of valorizing this vegetal input, thus ensuring a reasonable production cost and enhancing the meat quality [59].

### 4.4. Sunflower Oil

The Iberian pig is known for its high-quality meat, which is linked to its rearing in “Dehesas”, where they have access to acorns [2,32]. However, the supply of natural sources is seasonal, so Iberian pork products do not remain constant during the year. The addition of sunflower oil with high oleic acid content has been a popular approach in intensive Iberian pig production to simulate the fatty acid profile of pigs raised in traditional systems where acorns and pasture are available [2,26,32]. Studies have examined the effects of these diets on the fatty acid profile of different tissues, reporting inconsistent and tissue-specific effects. Benitez et al. [60] described that Iberian pig feeding with oleic acid had higher MUFA and oleic acid, lower SFA content, and a decrease in the n-6/n-3 ratio than the control group as a result of a major increase in the oleic acid diet which is consistent with the findings of previous authors [2,5,27,32]. These authors agree that adding high oleic sunflower oil is a successful approach for improving the attributes of raw meat, leading to enhanced cured products, which are the main sales of Iberian pork.

### 4.5. Linseed

Linseed is the ripe seed of flax, containing about one-third of oil, and more than 50% is α-linolenic acid (C18:3 n-3), making it an optimal ingredient in animal feed. Dietary flaxseeds enable C18:3 n-3 contained in muscle to compete more effectively with linoleic acid for the pathways involved in PUFA long-chain development. Using the whole seed could be more economical than the oil, and the whole seed contains natural antioxidants, which may slow down the oxidative effects of PUFA [29]. 

The results obtained by Mitchaothai et al. [61] indicate that the ratios of n-6/n-3 and C18:2 n-6/C18:3 n-3 in Kadon pigs could be enhanced by dietary supplementation of linseed oil (4.5%) without affecting performance production. These findings could be attributed to a reduction in the activity of Stearoyl-CoA 9-desaturase [29]. In addition, dietary supplementation with n3-PUFAs could positively affect other animal stages, such as pregnancy. The inclusion of PUFAS in the diet can prevent intrauterine growth restriction, increase birth weight, and reduce mortality. This is demonstrated in the study by Heras-Molina et al. [43], who described that a linseed diet in pregnant Iberian sows increases the PUFA proportion of fetal tissues, especially of n-3. These authors suggest that this ingredient diet could have a positive impact on the health status of piglets and meat characteristics.

### 4.6. Algae

In recent years, several investigations have been developed to use algae as an ingredient in animal feed. This is due to its nutritional composition, rich in high-quality proteins, antioxidant compounds, and PUFA content, such as C16:4 n-3, C18:2 n-6, C18:3 n-3, and C18:4 n-3. The benefits of algae as an ingredient are related to improving meat quality as n-3 is deposited and enhancing animal health due to bioactive compounds with immunological, growth-stimulating, or antimicrobial properties [62]. Nevertheless, the composition and subsequent benefits of using algae as animal feed could be affected depending on the specie evaluated. To our knowledge, no studies have assessed the effect of dietary algae on the fatty acid profile of meat from indigenous pigs. The study carried out by Altmann et al. [63] demonstrated that including *Spirulina sp*. in the diet of hybrid pigs showed higher levels of total n-6, C18:3 n-3, and γ-linolenic acid (C18:3 n-6) compared to the control group, which is in accordance with Costa et al. [62]. The inclusion of algae in the diet could probably positively affect pig fat in native pigs, as observed in hybrid pigs. Nevertheless, further research is needed on this topic due to the differences that could be shown depending on the breed assessed.

### 4.7. Insects

Insects are one of the dietary ingredients of free-ranging fish and monogastric animals worldwide, which have an interesting chemical composition, including protein, fats, minerals, and vitamins [46]. To the best of our knowledge, there have been no investigations about including insects as feed for indigenous pigs. Nevertheless, the studies conducted in hybrid pigs provide interesting results. Altmann et al. [63] described that *Hermetia illucens* dietary supplementation provides higher γ-linolenic and oleic contents in hybrid pork meat than the rest of the studied groups. In any case, results concerning the use of insects as part of pig diets are scarce. Considering the benefits reported in other monogastric animals [37], a deeper look into this topic could be interesting. In addition, insects could reduce organic waste; thus, using them as an ingredient would be consistent with rearing native pigs’ procedures.

## 5. Conclusions: Lessons and Perspective

Several factors could influence pork quality, such as farming system, nutrition, or animal breed. The studies referred to in this review indicate that there is a wide range of native pig breeds reared in different farming systems. It may be concluded that, in general, these breeds have a greater intramuscular fat content and improved lipid profile than industrial pork, enhancing the sensory, organoleptic, and nutritional quality of meat products. However, differences between breeds may be due to animal genetics, farming system, age or weight at slaughter, and muscle testing. Furthermore, nutrition is a well-researched approach to improving the fatty acid profile. There are numerous strategies, including the use of natural sources, that can be used for this purpose. Some, such as the use of acorns, sunflower oil, chestnuts, and olives, among others, have been extensively studied. Nevertheless, other strategies, such as insects and algae, have not been assessed in native pigs, and further research is needed. The studies carried out on industrial hybrids would be interesting to develop on native pigs, which would improve their breeding and the commercialization of the product.

## Figures and Tables

**Figure 1 animals-13-01696-f001:**
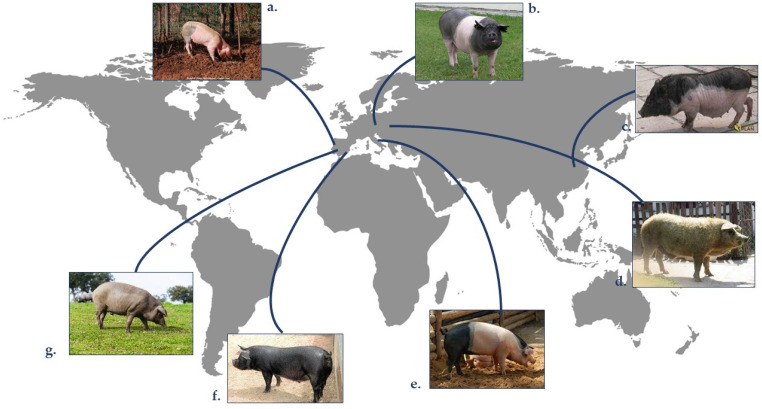
Geographical origin of some indigenous pig breeds. Generated by the authors. (**a**). Bísaro pig, (**b**). Prestice Black-Pied, (**c**). Wuzhishan pig, (**d**). Mangalica pig, (**e**). Cinta-senese pig, (**f**). Chato murciano, (**g**). Iberian pig.

**Table 1 animals-13-01696-t001:** A comparison of the fatty acid profile from different autochthonous and hybrid pig breeds.

Author	Authochonous Breed	Region	Hybrid	Fatty Acid Profile
Nevrkla et al. [9]	Prestice Black-Pied	Republic Czech	Large White × Landrace sows × Duroc × Pietrain boars	*M. longissimus thoracis et lumborum* - Decrease in C8:0, C10:0, C15:0, C22:0, C18:1 n-9, C18:3 n-6, C20:3 n-3, C20:4 n-6, C22:4 n-6, C22:5 n-3 and C22:6 n-3. - Increase in C18:1 n-9, C20:5 n-3, MUFA and MUFA/SFA. Back fat - Decrease in C10:0, C12:0, C14:0, C16:0, C14:1 n-5, C16:1 n-7, C18:1 n-7, C20:5 n-3 and C22:6 n-3 and SFA content. - Increase in C24:1 n-9, C18:2 n-3, C20:4 n-6, C22:5 n-3, PUFA, n-6 PUFA, n-3 PUFA, PUFA/SFA and MUFA/SFA.
Tomović et al. [36]	White Mangalica (M)	Hungaria	Large White (LW) and a crossbreed (LW × M)	*M. longissimus lumborum* - Decrease in C10:0, C11:0, C16:0, C17:0 and C22:0 Decrease in C15:1 c5, C17:1 t10 and C24:1. - Increase C16:1 t9, C16:1 c 9, C18:1 c9, C20:1 c11 and C18:2 c9,12. Increase in MUFA content with no differences in PUFA content.
Kušec et al. [28]	Black Slavonian	Croatia	(Pietrain × Duroc × Pietrain) × Camborough 23.	*M. longissimus thoracis et lumborum* - Decrease in C16:0, C18:0, SFA and PUFA n-6 content. - Increase C16:1 n-7, C18:1 n-9, C18:3 n-6 and C24:1 n-9, MUFA and PUFA n-3 content.
Serra et al. [21]	Cinta Senese	Northwest of Tuscany, Italy	Large White	Lard from backfat - Decrease in C16:0, C18:0, C22:5 n-3 and C22:5 n-3. - Increase in C16:1 c9, C18:1 c11, C18:1 c13. - Decrease in SFA and increase in MUFA and MUFA/SFA ratio.
Teixeira and Rodrigues [40]	Preto Alentejano	South of Portugal	Large White × Landrace	*M. longissimus thoracis et lumborum* - Increase in SFA (C16:0 and C18:0) and MUFA (C16:1 and C18:1) while inconclusive differences were found in PUFA.
Lebret et al. [41]	Basque	North of Spain	Large White	Subcutaneous adipose tissue - Lower SFA content due to the lower percentages of all SFA except C14:0. Lower proportion of PUFA resulting from lower proportions of C18:2 n-6, C18:3 n-3, C18:4 n-3, C20:3 n-6, and C22:5 n-3. - Greater MUFA proportion due to a higher proportion of C18:1 n-9.
Robina et al. [42]	Iberian	Spain	Duroc	Subcutaneous fat - Lower SFA and higher MUFA and PUFA values in pure Iberian.
Furman et al. [45]	Krškopolje	Slovenia	Maternal hybrid Large White × Slovenian Landrace (line 11) mated by Pietrain, Duroc, or Piertain × Slovenian Landrace (line 55) sires	*M. longissimus dorsi* - Decrease in C18:2 n-6 and C18:3 n-3, increase in C20:4 n-6, C20:5 n-3 and C22:5 n-3. Intramuscular fat content - Decrease in SFA content and PUFA content Subcutaneous adipose tissue - Increase in C18:1 n-9, C16:1 n-7, and C18:2 n-6. Decrease in C18:0 content. Krškopolje pigs had the highest MUFA, lower PUFA, and n-6 PUFA content.

## Data Availability

Not applicable.

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
