# Peer review of "Improving Fatty Acid Profile in Native Breed Pigs Using Dietary Strategies: A Review"

_animals, 2023, doi:10.3390/ani13101696_

Round 1
Reviewer 1 Report
The author compares the quality difference between the main domestic pork and commercial pork around the world and tries to improve the quality of pork from the perspective of nutrition. Overall, the topic is appropriate and the logical framework is clearer. The manuscript was suggested for publication in the Animals, with minor revisions. Specific comments/suggestions are as follows:
1. Line 124:Notice the punctuation after "Auqui et al".
2. Line 128:The commercial hybrids described here "Duroc, Yorkshire, and Landrace" refer to the "Duroc x Yorkshire x Landrace" triple cross or some other cros?.
3. Line 135:The necessary punctuation is missing between the two sentences.
4. Line 177:There should be a space between "<" and "0.5".
5. Line 264: Notice the punctuation after "Liotta et al"..
6. Line 284:Notice the punctuation after "Temperan et al".
7. Line 316:Title "3.3 Linseed" changed to "3.4 Linseed".
8. Line 333: The "n-3" described here refers to how many carbons are in.
9. Line 344:The secondary headings here should be the same as before, without bold font.
10. Line 277-327:Several natural resource components such as chestnuts, sunflowers and Linseed are introduced in animal feed to improve animal fatty acid profile. However, the effects of processed oil on pig fatty acid profile are only explained in the introduction of these resources.I suggest that the effects of processed sunflower seed meal and linseed meal on the fatty acid profile of pigs can be added.
no comments
Author Response
Dear Reviewer
Thank you very much for your kind comments
The comments made have been included in the manuscript in green color.
- Line 124:Notice the punctuation after "Auqui et al". It has been done (L182)
- Line 128:The commercial hybrids described here "Duroc, Yorkshire, and Landrace" refer to the "Duroc x Yorkshire x Landrace" triple cross or some other cros?.According to Huang et al. (2020). describe in their work the comparison with commercial hybrids (Duroc, Yorkshire and Landrace). Not from triple crossbreeding
- Line 135:The necessary punctuation is missing between the two sentences.It has been done
- Line 177:There should be a space between "<" and "0.5". It has been done (L235)
- Line 264: Notice the punctuation after "Liotta et al".. It has been done (L329)
- Line 284:Notice the punctuation after "Temperan et al". It has been done (L352)
- Line 316:Title "3.3 Linseed" changed to "3.4 Linseed". It has been done. L385
- Line 333: The "n-3" described here refers to how many carbons are in. Thanks reviewer
- Line 344:The secondary headings here should be the same as before, without bold font. It has been done
- Line 277-327:Several natural resource components such as chestnuts, sunflowers and Linseed are introduced in animal feed to improve animal fatty acid profile. However, the effects of processed oil on pig fatty acid profile are only explained in the introduction of these resources. I suggest that the effects of processed sunflower seed meal and linseed meal on the fatty acid profile of pigs can be added.
As the reviewer has dealt with, processed oil and meals from natural resources are widely used, but the aim of the work was the study the effect of natural resources, with minimum processing, particularly in native pigs breed. However, a sentence regarding this aspect has been added : (L102) In this regard, ingredients that require minimum treatment to be included in the diet of autochthonous pigs have been reviewed.

Reviewer 2 Report
The manuscript appears to be NOT systematic for the topic in the title of focusing the role of dietary strategies. There are excessive mentions to healthy aspects of native breeds fat which are not objectives of the manuscript.
The manuscript rises the clear differences observed on IM fat content among genotypes and rearing conditions in the literature. However, it ignores to quantify the likely role of diet on theses changes, and especially to give a mechanistic explanation to these differences. There is not any mention to the significant differences on the novo IM fatty acid synthesis among genotypes which merits a particular section (see for example different lipogenic enzymes activity between Landrace and Iberian pigs, Morales et al, 2003. Animal Science 77, 215-224.). Body fat is not only the consequence of absorbed dietary fat, there is also a significant synthesis of saturated and Monounsaturated fatty acids from glucose (mainly digested starch) that merits to be clearly described. There is a need to explain well the role of nutrients to enhance or modify the fatty profile of native breeds before moving to a description of potential ingredients.
Taking this into account may help to organize the description of likely ingredients. It is not the same to talk about acorns and chestnuts with a main content of carbohydrates than talking about the supply of sunflower oil or linseed oil. It is also important no to be ambiguous. The term "Sunflower" may also refer to sunflower meal. Then, it is better to talk about oils, and to explain the likely role of different oils composition to enhance or modify body fat (and IM composition). It is also intriguing that authors ignore acorns as an infredient for pigs.
These suggestion imply a new version of the manuscript whith a different focus and sections. Then, my suggestion is to recommend a new submission rather than extensively modifying the present manuscript.
Author Response
Dear Reviewer
Thank you very much for your kind comments
. We have considered that it is also important to include references to the characteristics of native breeds and why improving it would be interesting. In accordance with the reviewer's suggestions, a specific section called “Factors affecting IMF content and fatty acid profile” (in blue) has been included. Moreover, the reference suggested has been read and included, and new references to justify the mechanism of different ingredients to enhance the fatty acid profile have been added (in blue).
In this manuscript, we have tried to show the reader the different nutritional strategies to improve the fatty acid profile. For this purpose, we have tried to show a broad view of the different ingredients available (including oils or carbohydrate sources), whose studies have been carried out in native breeds. We have looked for ingredients whose transformation for inclusion in the diet is minimal. However, due to the importance of high oleic sunflower oil as an ingredient in breeds such as Iberian, it has also been included. The objective of the authors is to provide ingredients that can be included in the reuse of products while improving the fatty acid profile of the meat. However, it has not been an easy goal, since many of the ingredients have been studied in industrial pigs (hybrids). The authors thank the reviewer for the suggestion about acorns, which have been included. As they are the usual food of the Iberian pig, they had been forgotten. According to the reviewer’s recommendations changes have been made. Please, see the attached file. We have introduced changes to the manuscript and new sections that we hope will meet your needs.

Reviewer 3 Report
The premise to the review is very interesting. However, there are many grammatical errors, some of which are noted below, that need to be addressed. In addition, the authors need to provide more context in their review. They state that many different factors affect pork quality and that these factors vary in all of the different studies that are brought up, but they do not mention how these vary in this review. The goal of a review is put all of the information together for the reader. This reviewer thinks that some tables detailing this information and the many studies that are compared would add a lot of value to this manuscript.
More specificity in the simple summary – how many years? Why were they questioned? The authors also need a link between the first sentence and the second? Readers need to be able to track what is being said.
Line 40: supposing is a strange word here, I would suggest you rephrase
Line 41: the sentence beginning on this line is long and hard to follow
Line 56: combustive material?
Line 67: rusticity? Please clarify
Line 82: after the sentence beginning on this line, please add more detail on these studies.
Line 92: remove the word an
Line 101: IMF has been define before, be consistent throughout the paper
Line 118: the sentence beginning on this line needs reworded
Line 134: change included to including
Line 136: healthy tendencies? Health recommendation? From what organization?
Line 145: the data presented in this section would be better organized into a table perhaps.
Line 163: change to - Not only are the
Line 167: more validation/sources of the omega6:omega3 ratio is needed.
Line 172: the sentence beginning here needs a citation
Line 181: there is necessary to explain that – this phrase needs to be reworded
Line 255: in the sentence beginning here, try to re-word as it is grammatically incorrect
a fair bit of grammar editing is required
Author Response
The premise to the review is very interesting. However, there are many grammatical errors, some of which are noted below, that need to be addressed. In addition, the authors need to provide more context in their review. They state that many different factors affect pork quality and that these factors vary in all of the different studies that are brought up, but they do not mention how these vary in this review. The goal of a review is put all of the information together for the reader. This reviewer thinks that some tables detailing this information and the many studies that are compared would add a lot of value to this manuscript.
Dear Reviewer. Thank you very much for your kind comments. The authors have considered the comments and have incorporated the same in the manuscript. In addition, a section has been added to evaluate factors affecting fat content and fatty acid profile. The mechanism of action of the various ingredients that have the potential to modify the fatty acid profile has been included (in blue). The rest of recommendations have been incorporated in purple color
More specificity in the simple summary – how many years? Why were they questioned? The authors also need a link between the first sentence and the second? Readers need to be able to track what is being said. The following sentence has been added “Over the last decade, intensive pig-rearing systems have been questioned due to the farming method, leading to increased consumer demand for environmentally and animal-friendly pro-duction. These production systems, among other particularities, involve raising indigenous breeds that are widely appreciated for their rusticity and the high quality of their meat
Line 40: supposing is a strange word here, I would suggest you rephrase. Thanks reviewer. It has been rephrased “ Its consumption has a wide variety, as it can be eaten raw, cured or cooked
Line 41: the sentence beginning on this line is long and hard to follow: It has been replaced About 56% of the world's pork is produced in Asia, where China (with 48%) is the leading producer. This is followed by North America and the European Union, which represent 12% and 21% of world pork production. In Europe, Germany, Spain, and France are the main producers of pork
Line 56: combustive material?. It has been explained better: In addition, the intake of small shrubs and other vegetative species by indigenous pig breeds, combined with ruminant livestock, contributes to their removal and to the prevention of fire prevalence
Line 67: rusticity? Please clarify (L69) According to Bedia et al. (2012) the following information has been added: Many indigenous pig breeds are well-known for their rusticity. The term "rustic" is used to describe conventionally raised, slow-growing, local breed pigs, usually slaughtered at an advanced age, which provide carcasses and meat with high pigmentation of muscle and fat
Line 82: after the sentence beginning on this line, please add more detail on these studies. Dear reviewer. A separate section has been added (L103)
Line 92: remove the word an: It has been done
Line 101: IMF has been define before, be consistent throughout the paper. It has been replaced by IMF
Line 118: the sentence beginning on this line needs reworded: It has been done. (L176) These results are probably due to the genetics of the native breeds, which have been linked with changes in fat absorption and deposition
Line 134: change included to including. (L192) It has been done
Line 136: healthy tendencies? Health recommendation? From what organization? (L194) The following reference has been included: “EFSA Panel on Dietetic Products, Nutrition, and Allergies (NDA); Scientific Opinion on Dietary Reference Values for fats, including saturated fatty acids, polyunsaturated fatty acids, monounsaturated fatty acids, trans fatty acids, and cholesterol. EFSA Journal 2010; 8( 3):1461. [107 pp.]. doi:10.2903/j.efsa.2010.1461”
Line 145: the data presented in this section would perhaps be better organized into a table. It has been added. See the Table 1. Thanks (L238)
Line 163: change to - Not only are the. It has been done
Line 167: more validation/sources of the omega6:omega3 ratio is needed. References have been added
Line 172: the sentence beginning here needs a citation. It has been done
Line 181: there is necessary to explain that – this phrase needs to be reworded. It has been delete.
Line 255: in the sentence beginning here, try to re-word as it is grammatically incorrect.(L321) It has been done. In the study conducted by Leite et al. [19] with Bisaro pigs fed with crude olive oil, an increase in IMF was observed with respect to the control group,

Round 2
Reviewer 2 Report
Authors did a great job in the response to report 1 and the new version of the manuscript. It was not acceptable to have a review about factors affecting fat composition in the meat of porks (including intramuscular fat) and not describe or include the inputs about the endogenous synthesis or the role of acorns for iberian pigs.
The manuscript is now systematic about the mechanisms, and also show a detailed piece of work of sources that can be used as alternative for feeding pigs.
Reviewer 3 Report
The authors have done a good job in addressing all of my previous concerns. I believe that the papers flows a lot better and goes into an appropriate level of detail now.
The manuscript still requires minor to moderate editing of the English language.